



# Proposed Best Practices for Collaboration at Cross-disciplinary Observatories

Jason Philip Kaye[1], Susan Louise Brantley[2,3], and Jennifer Zan Williams[3] and the SSHCZO team*

[1]Department of Ecosystem Science and Management, The Pennsylvania State University, University Park, PA 16802 U.S.A.
[2]Department of Geosciences, The Pennsylvania State University, University Park, PA 16802 U.S.A.
[3]Earth and Environmental Systems Institute, The Pennsylvania State University, University Park, PA 16802 U.S.A.
* A full list of authors and their affiliations appears at the end of the paper.

*Correspondence to*: Jason Kaye (jpk12@psu.edu)








**Abstract.** Interdisciplinary science affords new opportunities but also presents new challenges for biogeosciences collaboration. Since 2007, we have conducted site-based interdisciplinary research in central PA, USA at the Susquehanna Shale Hills Critical Zone Observatory. Early in our collaboration, we realized the need for some best practices that could guide our project team. While we found some guidelines for determining authorship on papers, we found fewer guidelines describing how to collaboratively establish field sites, share instrumentation, share model code, and share data. Thus, we worked as a team to develop a best practices document that is presented here. While this work is based on one large team project, we think many of the themes are universal and we present our example to provide a building block for improving the function of interdisciplinary biogeoscience science teams.

# 1 Introduction

Interdisciplinary science has proliferated in recent decades, resulting in larger science teams drawing on increasingly complex research infrastructure (Lattuca 2001, Rhoten and Parker 2004, U.S. National Research Council 2005, Pearce et al. 2010, Hinkley et al. 2016). The scientific community has embraced interdisciplinary research because it leads to discoveries that would not arise from work in isolated disciplines and because many of the most intriguing questions lie at the boundaries of fields. However, science that crosses disciplines and is generated by teams also brings new challenges related to attribution of credit and management of shared infrastructure and data. While social scientists have studied interdisciplinary science (Lattuca 2001, Rhoten and Parker 2004), domain scientists themselves have not thoroughly grappled with the problems that arise. As a result, there are relatively few published models or guidelines to promote efficient and collegial management of interdisciplinary science. While guidelines for co-authorship have been published (Weltzin et al. 2006, Oliver et al. 2016), this represents just one of the challenges of interdisciplinary collaboration. Other aspects, such as shared equipment, co-location of equipment, maintenance, sharing samples and computer codes, etc. are all areas where we lack established guidelines to promote efficient collaboration.

Interdisciplinary work in the biogeosciences has been catalyzed recently by the emergence of the Critical Zone (CZ) approach. Earth's CZ is the thin near-surface zone spanning from bedrock to the atmospheric boundary layer (U.S. National Research Council 2001, Brantley et al. 2007). Since the mid-2000s, scientists have been viewing this zone through a new interdisciplinary lens that brings together biology, soil science, geology, hydrology, and meteorology to make co-located measurements of chemical and biological transport and transformation that describe past landscape evolution and improve projections of future conditions (Brantley et al. 2016, Sullivan et al. 2019). This interdisciplinary approach has precipitated important insights that link hydrology, weathering rates, soil characteristics, nutrient availability, microbial process, and plant dynamics (e.g. Hahm et al. 2014, Richter et al. 2015).

While interdisciplinary CZO research is attracting great scientific interest and funding opportunities, it also creates problems that most scientists are not trained to address. Disciplinary norms vary in how credit is attributed for publications, data, and model code. In addition, the CZ approach relies on a wide array of instrumentation that must be deployed systematically, and





again, disciplinary norms vary in how to prioritize instrumentation siting and maintenance. To our knowledge, there are no

published guidelines to facilitate collegial and efficient management of CZ science. In this paper, we seek to initiate a
community-wide discussion about CZ collaboration by sharing our own guide for "best management practices" at our Critical
Zone Observatory (CZO).

The Susquehanna Shale Hills Critical Zone Observatory


In 2007, the US National Science Foundation began funding a network of CZOs (Brantley et al. 2017), and one of these initial
CZOs was the Susquehanna-Shale Hills Critical Zone Observatory (SSH CZO), where we work. After collaborating for about
a decade and discussing ideas in several all-team meetings, we distilled lessons learned from our successes and mistakes into
the best practices guide presented below. Ours is not a definitive model, but rather a single and evolving example that is meant

to foster discussion of how to maximize the benefits of interdisciplinary science. While section I (co-authorship) has been
addressed in some prior publications (Weltzin et al. 2006, Oliver et al. 2018), the major new contributions of this document
are the treatment of other aspects of interdisciplinary science including managing infrastructure, advising students, and sharing
model codes and data. Our format is to first lay out the core concepts for key themes (sections I, II, III, etc), and then use a
litany of questions and answers to flesh out details for important cases that arise over time. We implement this document by

asking CZO scientists to discuss it and sign it in periodic team sessions.

There are some critical components of any team running an environmental observatory that need to be defined. Different
observatories may define their core staff in different fashions, but all such observatories must accomplish similar tasks. For
example, typically a Program Coordinator coordinates the personnel and the reporting and team meetings.  In addition, our

Program Coordinator is in charge of registering and archiving samples. Typically, a Data Manager is responsible for managing
data that streams from the catchment as well as publication of data from other instrumentation online. Finally, a Watershed
Specialist is needed to manage the field deployment of instrumentation, including the data streaming. This person also works
with the Program Coordinator to balance competing needs for space or instrumentation or conflicts in watershed usage.
Similarly, the Watershed Specialist and Program Coordinator must coordinate the team in maintaining the site clean and

orderly.  A Director supervises all three of these personnel.

The governing body of the SSH CZO is a Steering Committee described in Section IV. The home institution for our team is
the Pennsylvania State University (abbreviated "Penn State" below). Our CZO includes one large watershed, Shavers Creek,
as well as three nested subcatchments, each with distinct ownership and permitting. Shavers Creek watershed in entirety is 165

km$^2$ and is owned by many individuals – our work throughout the catchment is limited to public lands or lands where we have
specific permissions. The Shale Hills subcatchment (Brantley et al. 2018) is part of the Penn State Sustainable Forest and all
permitting occurs through the Penn State Forest Lands Office. The Garner Run subcatchment is owned by the State of



Pennsylvania and permitting occurs through the Pennsylvania Department of Conservation and Natural Resources (Brantley et al. 2017). The Cole Farm subcatchment is privately owned and access is granted as part of an agreement with the landowner (Li et al. 2018).

## 2 Best Practices Document

### 2.1 Best Practices for Authorship on Peer Reviewed Papers

Our criteria for authorship are based the Ecological Society of America code of ethics (ESA 2013) and they are consistent with recommendations of the American Geophysical Union Committee on Publication Ethics (Albert and Wagner 2003). Authorship may be anticipated if researchers make substantial contributions in one or more of the following areas:

1) creation of the conceptual ideas or experimental design;

2) management or execution of the study;

3) analysis or interpretation of data; or

4) writing of the manuscript.

We do not prescribe levels for substantial contribution, and so each manuscript will require an open discussion regarding authorship. However, to provide some guidance, substantial is taken here to mean a contribution that either involves planning and analysis beyond that available at a commercial laboratory, creative or long-term field work, development of models, or other similar contributions. In general, engagement in writing is often a key delineation of co-authorship. Thus, it is important that scientists contributing to #2 (e.g. long-term collection of field data) are sought out and afforded the opportunity to contribute in the analysis and writing stages of the manuscript. It must be recognized that different disciplines have different codes of authorship and so flexibility must be retained. Regardless, the discussion and agreement should be achieved early in the collaboration and the senior scientist should promote this discussion. In ambiguous cases, we are inclined to err on the side of being more generous with authorship. Once established, authorship and the order of authors shall not be changed without consulting all the authors on the manuscript. No author shall be included on a manuscript that has not agreed to the content in the final version. This means that every author must be given a reasonable amount of time to read revisions of the manuscript, but, in turn, if an author does not respond for revisions in a reasonable amount of time, they can also lose co-authorship.

Some questions that have arisen are discussed specifically below.

### 2.1.1    If I use someone's old, published data, should they be included as a co-author?

No, prior publication of data does not, in itself, constitute a significant contribution to new papers.



### 2.1.2    If I use someone's old, unpublished data should they be a co-author?

If the data are unpublished but also not embargoed, then we encourage the authors to engage the scientist who collected the
data at a level that would constitute a substantial contribution. However, if a good faith effort is made to engage the scientist
in charge of the original data and that scientist has not responded, then it would not be appropriate to include them as a co-
author (but it would be appropriate to acknowledge them). If the old data are embargoed (i.e., not yet public) then the authors
must gain permission to use the data. At this time, the two parties (paper authors and data collector) should discuss authorship
in the context of the criteria described above. In unusual cases, a researcher who collected embargoed data may not make
appropriate progress in publishing a dataset. In that case, the CZO team may need to decide on a course of action with respect
to publication of the embargoed data that, in the best case would involve discussion with the original researcher but might have
to proceed without such discussion. Such unusual circumstances should be well discussed among the steering committee for
guidance. Ultimately, a researcher who makes a substantial contribution to a manuscript should be included as a co-author on
a publication.

### 2.1.3    If I use someone's code or model output from a previously published paper, should they be included as a co-author?

No, unless the code developer is intellectually engaged in the manuscript development. A couple of examples that might lead
to authorship: 1) The code developer provides new model outputs and is engaged in output analysis; 2) the code developer
runs new model simulations for the manuscript (i.e., performs new calibration, collects new driver data), or adds new
functionalities to the model.

### 2.1.4    If I use someone's code that has not been published in a paper, should they be included as a co-author?

Similar to using someone's unpublished data, we encourage the authors to engage the code developer at a level that would
constitute a substantial contribution.

### 2.1.5    If I collect field samples for someone should I expect to be a co-author on their paper?

Field sampling is often an overlooked component of the creative scientific process where critical decisions are made that affect
the quality and value of the data. However, field sampling alone is not a contribution that automatically warrants co-authorship.
We encourage discussions that enable people who have contributed substantially to field work to become engaged in analysis
and writing at a level that warrants co-authorship. The long-term nature or difficulty of field collection can also be taken into
account.



### 2.1.6    If I test an idea from a CZO proposal, should the Principal Investigators (PIs) be co-authors on the paper?

This is a tricky question and varies from one team to another. For example, in some observatory teams, every paper that is published includes the name of the Principal Investigator. At the SSH CZO, the answer to this question depends on how specific the idea is and how much input the PI has had on the project and the paper. If the authors of the proposal conceived of the idea and described an experimental design to test it, then they may have met criterion #1 for co-authorship, and they should be given the opportunity to meet other criteria for co-authorship. On the other hand, at our CZO, if the research is not tied to hypotheses that are described in the proposal, then the proposal PIs should not be included as authors simply because they were a PI on the proposal. In addition, PIs may not have generated every hypothesis in the proposal: some work that is accomplished may thus not warrant PI authorship.

### 2.1.7    If an undergraduate researcher collected some of the data, should they be a co-author?

Undergraduate researchers should be considered for authorship under the same criteria as other scientists. We should promote co-authorship in this regard by giving research interns opportunities to contribute to data analysis and writing if the student is ready for such efforts and remains with the team for a sufficient amount of time. However, in some cases, a worker may only do "what is told" and not participate in planning or thinking about the results in any substantial way: in these cases, inclusion as a co-author may not be warranted.

### 2.1.8    How long should co-authors have to review a manuscript?

Co-authors should discuss timelines for each manuscript. However, a reasonable expectation is that co-authors will read a draft within one month of receiving it, assuming that the author has established some sort of reasonable timeline with respect to vacations, trips, etc. Shorter turnaround times may be appropriate for revisions, but co-authors are still expected to read the final (revised) version.

### 2.1.9    What do I do if I try repeatedly but I cannot get a co-author to read the manuscript?

An appropriate approach is the following. When the author finishes a version of the manuscript, he or she discusses with the possible co-authors a timeline or sequence of review (in other words, the authors must have some ability to frame up the timeline – it is not just at the discretion of the first author). If a potential co-author does not read or comment appropriately on a manuscript, the author should propose a reasonable deadline and write in an email, "we will submit this paper without your name unless you read it and comment on it by such and such date: we prefer to retain you as co-author but we must move forward". In case the potential co-author still does not respond, it is appropriate to remove the potential co-author from the authorship list. All attempts should be made for other authors to contact the co-author by multiple means (e.g. email and phone) and make it clear that they will be removed from the authorship list if they do not respond in a specified amount of time. One possibility is also to submit a paper without a co-author (because the co-author cannot participate in paper writing at the time)





and then add the co-author back in later if they re-engage appropriately and it is cleared appropriately with the journal editorial board.

### 2.1.10    Who will decide the final author list in cases of contention?

We expect co-authors to handle this problem in a collegial way. Best practice will always dictate that the discussion of co-authorship be initiated early in the process and be continued throughout the process. The senior scientist on each project should
guide this process along. Guidance can also always be sought from the CZO Steering Committee and the Director of the Observatory.

### 2.1.11    How is the order of authors determined?

Best practice would be for all the co-authors to decide this in a collegial way; in most cases, the senior author will decide the order of authors. Order of authors is particularly sticky in some cases because different disciplines view author order
differently. On the other hand, these differences can also lead to easy choices. For example, in most disciplines first authorship is the most highly regarded position; however, in chemical sciences the senior author is often listed last and that is considered a prestigious position as well (Sauermann and Haeussler, 2017).  In general, the person who frames and writes the paper should be first author.

### 2.1.12    Who should be the corresponding author on a paper?

It has been our experience that even larger differences in opinion are present among scientists from different disciplines with respect to corresponding author. To some scientists, the corresponding author is simply the lead author of the paper. To others, the corresponding author should be the author who conceived the project, procured funding for the project, and is in a stable career position and would be most likely to be easy to reach for future correspondence. Often, the lead author may be unwilling, unprepared, or unavailable to field questions from the journal and future readers of the paper and it may be appropriate to
assign a co-author to be the corresponding author. To some scientists, it is considered excellent training for PhD students to be corresponding authors on papers when they are the lead author. The question of assignment of corresponding author is also of note in that for some junior scientists from other countries, this assignment carries great weight. Best practices here must again rely on engagement and conversation early in the planning of the paper.

### 2.1.13    How can we remember to include all the appropriate co-authors?

In highly interdisciplinary and large teams, it is not uncommon that an author prepares a paper and forgets to include appropriate co-authors that made significant contributions early in the project. This has happened several times at the SSH CZO and led us to institute a policy whereby every authorship team that starts to put together a paper is asked to share the proposed title, topic, and author list with the Program Coordinator early in the writing process. The Program Coordinator then shares the information with the observatory director and an email is sent out to the rest of the team asking if anyone thinks that





they should be on the paper as a co-author or if they think they have a significant contribution to make to the paper. Again, discussion can then ensue to decide on authorship and order.

### 2.1.14   Does everyone in the team have to agree with everything that is written in every paper?

Again, this can be a tricky problem in interdisciplinary science. In general, we have experienced many instances wherein project members did not agree on interpretations of data: amicable collaborations were nonetheless pursued and papers
published. We encourage ample discussion among the team to learn from one another in such cases. In some rarer cases, co-authors may not entirely agree with every interpretation in a paper; however, the senior author should make every attempt to promote discussion and language that can be agreed upon by the authorship team in the publication.

### 2.2 Best Practices for Installing Infrastructure or Experiments

Best practices for installation of infrastructure require not only careful consideration of impacts on the environment but also
on existing infrastructure and needs of other team members. Installation must also take into account the permitted use of sites and usage fees. Scientists (including CZO PIs) that would like to initiate new work that is co-located within the bounds of the CZO must propose each idea for installation with the CZO Steering Committee, the Program Coordinator, and the Watershed Specialist, and typically each installation is described for the entire team for comment. As scientists outside the initial team begin to propose work in the site, the Observatory Director identifies key CZO scientists who must be consulted regarding the
new project. PIs are encouraged to share the information with all students in the lab group so potential impacts can be considered. A second email should be sent prior to the installation of the new equipment. If the new research includes destructive sampling or activity that could affect many projects, then the Steering Committee might present the proposed work in an all-hands meeting to discuss the viability of the new project. If there are conflicting deployments, then the Steering Committee has the responsibility to determine whether new installations should go forward.


The Watershed Specialist should be included in both preliminary and developing conversations regarding new equipment. The final placement of all new field infrastructure (e.g. sensors, pvc, etc) must be approved by the Watershed Specialist. In addition, materials that will stay in the field are marked with a PI-specific color using paint, tape, flagging, or some other permanent coloring. Metal tags stamped with identification are often used. Even non-Penn State or non-NSF personnel are assigned a
specific color and are expected to maintain their color coding while working in the project. Color coding is managed directly with the Watershed Specialist and the Program Coordinator.  Immediately after installation of new equipment it is a best practice to take a photograph of the installation and share it, by email, with the entire team. In addition, the location of the instrument must be communicated to the Data Manager who can update maps of equipment.

The current usage agreement with Penn State Forest Lands Office allows CZO top-tier priority research within the Shale Hills catchment. Any outside funded project must be approved by the CZO Steering Committee, followed by approval by the Penn



State Forest Lands Office. The Forest Lands Office may impose a separate research permit and usage contract and usage fee. Projects are generally not considered to be under the CZO umbrella by default although they may eventually be placed there, with the exception of seed grant projects funded by the SSHCZO.


As the CZO expands outside of Penn State lands, new rules are being developed. Specifically, the CZO has an agreement with the Pennsylvania Department of Conservation of Natural Resources (PA DCNR) for specific activities in the Garner Run watershed. Every person who works in that area as part of the CZO (student or faculty, inside or outside of Penn State) and every advisor for a student working at the CZO on the specific activities described must sign the agreement with the PA DCNR

and this must be kept on record by the CZO Program Coordinator. If a PI initiates new work in the area that is not listed in our permit, a new permit must be requested and negotiated directly between the PI and the DCNR, and a record of this documentation must be kept on hand by the Program Coordinator. It can take up to 3 months for the permit process with the DCNR. If work is pursued in these lands without signing the form, or if work is pursued which is not described on the agreements, the CZO will rescind permission to work on the project and will work with DCNR to rectify the situation. As we

expand to private lands, additional guidelines will be developed, and extra care will be taken to respect the wishes of the land owners.

In some cases, observatories may include private land or land enabling specific land use practices. For example, our observatory work has recently grown to include one subcatchment on a practicing farm (Li et al. 2018). In this case, the

Watershed Specialist has been designated as the point person for all communication with the landowner, and the Watershed Specialist works closely with the land owner and farmer (two separate people) so that observatory activities do not disrupt the functioning if the farm. Likewise, in sampling of the mainstem of the stream throughout the watershed, every CZO worker only accesses public land, or asks for permissions to step on private land before sampling. Some private landowners have refused permission for access, and this lack of access is strictly observed. One benefit of working on private land is that CZO

workers can sometimes interact with the landowner and farmer, and every attempt has been made to learn from them as well as to give them information in return. As part of these efforts, the CZO team also works with extension agents through Penn State.

New research that is not co-located with existing CZO infrastructure may require a revision of the CZO permit and will need

to be discussed with the Forest Lands Management Office, the DCNR, or the Cole Farm landowner. The CZO Project Coordinator and Watershed Specialist should be included in these discussions. In general, best practice will initiate discussions with the CZO Steering Committee, followed by discussions with the landowner. When new funding is garnered for new research at the CZO, a new fee will generally be paid to the Penn State Forester for this work. This fee will be negotiated directly with the Forester.






When a PI receives new funding for new instrumentation (separate from the CZO grant), the CZO itself will not become responsible for the new infrastructure that is emplaced in the CZO catchments. Likewise, the new PI will be encouraged to use the CZO's data infrastructure for publication of data; however, the CZO will not become responsible for the data from the new project nor will the CZO police publication of the new data. Ultimately however it is recognized that the PI is generally co-
locating the experiment at a CZO catchment due to the pre-existing research and infrastructure. Given this "value added" by the CZO, the CZO Steering Committee and Watershed Specialist can ultimately decide whether certain activities are pursued in the CZO catchments. For example, a proposal might be funded to do geophysical monitoring in Shale Hills and might involve a new permitting fee to the landowner. After initiation of the work, the PI of the new proposal might decide he/she wants to do trenching up the middle of the catchment. If the CZO Steering Committee decides this is inappropriate, then the
new PI will not be enabled to complete the trenching. In this regard, the Steering Committee will work closely with the landowner to maintain appropriate activity.

### 2.3 Best Practices for Using, Maintaining, and Sharing Existing Field Infrastructure

All infrastructure at the CZO is linked to a PI via color coding (see section II). This PI is responsible for maintaining and promoting collaborative use of the equipment. While the color codes denote the PI in charge, they do not denote ownership of
equipment. All CZO field infrastructure and data are shared. However, no field equipment should be used without first notifying the PI in charge and establishing the terms of use and collaboration. Shared use and collaboration is expected and in some cases, this may mean developing a plan of collaboration that could lead to co-authorship if criteria in Section I are met. If PIs cannot agree on terms of shared use, then they should bring the issue to the Steering Committee.

The PI in charge may decide that it is best not to maintain equipment in working order, even though the equipment can remain in the field for future activities. For example, lysimeters can stay in place for years without being sampled. In these cases, the PI in charge should notify the Watershed Specialist and any co-PIs that have used the equipment in the past. A new PI may want to initiate the use of that instrumentation. In that case, the new PI and the original PI will be considered in charge of the equipment and its use. Any time infrastructure is moved or removed, the person in charge should contact the Data Manager to
report the equipment, PI, geolocation data, and the date of change.

While shared use is the overarching goal, there may be some equipment for which shared use is not appropriate. For example, some cases might involve equipment which is very sensitive or difficult to maintain or expensive or rented or borrowed. These can be handled on a case by case basis.


Questions that may arise:



### 2.3.1    What if I cannot maintain the equipment myself?

There are cases in which the CZO support staff or collaboration among co-PIs may be required to maintain field infrastructure. These will need to be handled on a case by case basis with consideration of the availability of support staff time. In general, 330 when a PI begins a sub-project that will require time from support staff, that requirement must be vetted through the Steering Committee. The Watershed Specialist will generally be the person to help in maintenance.

### 2.3.2    What if an investigator is not maintaining critical equipment in a way that promotes shared use?

In these cases a broader discussion may be needed in which the team may decide to transfer maintenance responsibilities to a different investigator or to allocate more project resources (support staff time or funds for maintenance) to the equipment.

## 2.4 Best Practices for Removing Field Infrastructure

If field infrastructure has reached the end of its useful life it should be removed by the PI in charge, as denoted by the color coding, and the landscape returned to original form. There may also be cases in which the equipment is still functional, but the PI wants to remove the equipment to reduce the maintenance burden. Before removing equipment for any reason, the PI should work with the Watershed Specialist to email the CZO team (all co-PIs plus support staff) to determine whether the removal 340 will affect other users.

When the CZO ceases to be a continuing research project, or when a sub-project ends, each PI has the responsibility to remove equipment with their color code or negotiate a new use agreement with the landowner. Our current CZO use agreements stipulate that we will restore the landscape to a pristine condition when we are finished with the project. Each year we also host a watershed cleanup day to pick up litter and maintain the infrastructure.

## 2.5 Best Practices for Collecting, Sharing, and Archiving Samples

Before going to the field to collect samples, scientists should make the Watershed Specialist and/or Program Coordinator aware of their sampling schedule. This is typically done via quarterly planning that is solicited by email. Sampling protocols should be posted on the CZO shared data space (a specific storage space should be defined here) so that all future users can use the same sampling protocol or deviate intentionally. CZO workers should attempt to share samples so that multiple analyses 350 are conducted on the same sample. The scientists sharing the samples should agree on the terms of the collaboration, including the potential for co-authorship.

Every solid and liquid sample collected from the field should become archived if sufficient sample is available and if it is likely or possible that future users might want to access this sample. The Program Coordinator is responsible for sample 355 archiving. PIs and their students and postdocs should consult with the Program Coordinator prior to collecting any samples so that the archive protocol can be established. The CZO has an established location for dry storage for solid and water samples.



No archive is available for frozen samples. All samples must be registered with International Geo Sample Number (IGSN) (http://www.geosamples.org/igsnabout) prior to archival. CZO personnel should attempt to share archived samples with one another and with the broader scientific community. Scientists who want to use archived samples are required to contact the

Principal Investigator and describe how the sample will be used. The Program Coordinator is responsible for facilitating this communication and sharing. Often it is best to discuss the terms of collaboration before the archived sample is released. However, in cases when the collector cannot be consulted or does not consent to the release, the case can go the Steering Committee. If archive sample retrieval becomes overly time-consuming, arrangements may need to be made to pay someone to find samples.


Questions that may arise:

### 2.5.1    What if I want to deviate from the established CZO sampling protocol?

We expect this to happen. A rationale should be provided for the change and methodologies should be noted in protocols maintained in the shared data storage space so that others will know how and why the change was made. The Program

Coordinator will facilitate and oversee modifications to the protocols.

### 2.5.2    What if there is only a little bit of an archived sample left and someone wants to use it up?

If the collector and user of the archived sample and PI of the CZO agree that this is a good use of the sample, then it can be used. In general, however, samples should not be used up. If there is disagreement, then the Steering Committee can be

consulted.

### 2.6 Best Practices for Sharing Data

Guidelines for sharing CZO data are outlined here: http://criticalzone.org/national/data/access-czo-data-1national/#DataUseAgreement. Where possible, a PI should get a DOI for datasets for future citation. In general, we consider that data storage in the CZO data infrastructure is advisable, even for data funded by entities outside of Penn State

NSF CZO funds. However, the CZO does not become responsible for archiving these data.

It is a best practice not to directly share your copies of data with third parties. For example, if you have an excel spreadsheet of data that another student or PI has shared with you, you should not share those data with a third scientist. Instead, it is best to have that scientist access the data by going directly to the CZO web page or contacting the original data source (PI and

student) directly. Under some circumstances (e.g. when you have manipulated data in a way that is beneficial to the third party)





you may need to pass on someone else's data to a third party, you should obtain written consent from the original data source, for example through an email exchange that includes a discussion of terms of authorship and use.

Some data sharing will occur prior to uploading the data to the CZO database. Data sharing at this early stage is encouraged and even necessary to enable students and PIs to conduct multidisciplinary research. The parties involved should establish authorship and use expectations at the time the data are shared. As discussed above, data should never be shared with a third party without first consulting and obtaining written consent from the original source of the data.

## 2.7 Best Practices for Project Management

The Steering Committee shall be comprised of a subset of the PIs (some fixed and at least one rotating), a subset of the staff,
and one rotating student. The Steering Committee should send out updates after their meetings to keep co-PIs appraised of key decisions. The Steering Committee is an appropriate outlet for all grievances related to the project. Discussions of sensitive issues (e.g. personnel) need not be shared, but decisions regarding allocation of resources and discussions about important changes affecting PIs should be shared.

As new PIs become involved in the CZO, the Steering Committee and all of the PIs will make every attempt to avoid the situation where more than one group is working on the same problem. However, some overlap will undoubtedly happen, and some overlap is expected to be appropriate in some cases. The Steering committee will thus try to steer PIs toward collaborative approaches to overlap, or toward appropriate "competition". In this regard, "competition" means collegial testing of alternate hypotheses or alternate methodologies to understand functioning of the CZO. The CZO management ultimately has no
authority to prohibit publication of ideas, data, or models for the CZO and in fact encourages competing ideas, data, and models.

In general, the CZO management will make every attempt to promote i) collegiality, ii) open communication, iii) excellence in research, iv) excellence in education, v) excellence in collaborative science, vi) excellence in outreach to the public.

A field crew comprised of a rotating group of students, postdocs, and staff supported by the project will assist with sample collection and general maintenance at the site and will help ensure that field sampling can always be conducted in pairs.

## 2.8 Best Practices for Advising Students

In general, graduate and postdoctoral students who work at the CZO should be encouraged to appear as co-authors on joint
publications as appropriate. Generally, a student will be first author on the project they spearhead (if they do most of the work), unless they do not move forward on publication in a timely manner. When students do not move forward on a project within one year of completion of their degree, the PI may write the paper and first author the project.



It is the responsibility of PIs on the CZO to mentor their students regarding CZO best practices. Having a student sign this
document is not enough; continuous mentoring regarding ethics and best practices is expected. PIs are expected to be aware
of which data and models their students are using, which datasets originated from other CZO students or PIs, and to be engaged
in all discussions regarding authorship and use of data, models, and infrastructure. Furthermore, PIs are expected to share
relevant emails with their students including those related to infrastructure and site maintenance.

## 2.9 Best Practices for Outreach

The CZO has a commitment to complete outreach to non-scientists and the public in general. It is expected that everyone who
works at the CZO will at some time (e.g. once per year) participate in public outreach coordinated by the CZO. However,
appropriate clearances are often required before PSU faculty and staff can participate in outreach with certain populations (e.g.
under-age students).

## 2.10 Best Practices for Reporting

It is expected that everyone working at the CZO will provide reports of effort to the Program Coordinator in a timely manner.
Lack of participation in reporting, if egregious, can be grounds for termination of collaboration at the CZO. Everyone working
at the CZO will also be expected to cite the CZO appropriately (as indicated on the CZO website) and to provide copies of
submitted, in press, and published papers to the Program Coordinator at the time of submission, acceptance for publication, or
publication respectively.

## 3 Conclusion


There is growing evidence that collaborative teams advance science in distinct ways from individual investigators (Uzzi et al.
2013). This may occur because each PI brings deep, but often conventional, understanding of their knowledge domain into
innovative combinations with collaborators from other domains. Teams can only leverage these innovative ideas if they work
well together through collegial and efficient use of field sites, instrumentation, samples, data, and model code. One key step is
agreeing on the best practices for working together. Once per year at a project meeting, we discuss best practices, as outlined
above, with the entire team of the Susquehanna-Shale Hills Critical Zone Observatory. These discussions typically bring to
light challenges that are then added to our living best practices document. Most senior scientists at our CZO were not taught
to work in teams, developing this best practices document has helped experienced and young scientists alike to grow an
understanding for efficient ways to collaborate.  We offer this document as one example, with the hope that it will foster
discussion enabling the field of biogeosciences to fully capitalize on large-team collaborative science.





**Author Contribution**

J.P.K. wrote the initial draft and is the principle and corresponding author of this living document built from ideas generated and conversations had throughout this collaborative project. S.L.B. conceived of the idea of publishing this best practices document and contributed to writing sections. J.Z.W. provided logistical leadership and support in the implementation and documentation of these best practices, including soliciting and synthesizing team input. The team provided insightful contributions to the challenges experienced and ideas for practical application of resolution.


**Team List**

D. Eissenstat (Dept. of Ecosystem Science and Management, Penn State University, USA), K. Davis (Dept. of Meteorology and Atmospheric Science, Penn State University, USA), L. Li (Dept. of Civil and Environmental Engineering, Penn State University, USA), T. Russo (Dept. of Geosciences, Penn State University, USA), R. DiBiase (Dept. of Geosciences, Penn
State University), H. Lin (Dept. of Ecosystem Science and Management, Penn State University, USA), M. Kaye (Dept. of Ecosystem Science and Management, Penn State University, USA), Y. Shi (Dept. of Ecosystem Science and Management, Penn State University, USA), L. Guo (Dept. of Ecosystem Science and Management, Penn State University, USA), E. Hasenmueller (Dept. of Ecosystem Science and Management, Penn State University, USA), K. Brubaker (Environmental Studies, Hobart and William Smith Colleges, USA), T .Adams (Dept. of Ecosystem Science and Management, Penn State
University, USA), C. Bao (Dept. of Energy and Mineral Engineering, Penn State University, USA), J. Del Vecchio (Dept. of Geosciences, Penn State University, USA), X. Gu (Dept. of Geosciences, Penn State University, USA), Y. He (Dept. of Meteorology and Atmospheric Science, Penn State University, USA), B. Hoagland (Dept. of Geosciences, Penn State University, USA), W. Reed (Dept. of Ecosystem Science and Management, Penn State University, USA), I. Szink (Dept. of Ecosystem Science and Management, Penn State University, USA), J. Weitzman (Dept. of Ecosystem Science and
Management, Penn State University, USA), D. Xiao (Dept. of Energy and Mineral Engineering, Penn State University, USA), B. Forsythe (Earth and Environmental Systems Institute, Penn State University, USA), B. Dillner (Dept. of Ecosystem Science and Management, Penn State University, USA), C. Hodges (Dept. of Ecosystem Science and Management, Penn State University, USA), V. Marcon (Dept. of Geosciences, Penn State University, USA), E. Primka IV (Dept. of Ecosystem Science and Management, Penn State University, USA), P. Silverhart (Dept. of Geosciences, Penn State University, USA), and Q.
Tang (Dept. of Ecosystem Science and Management, Penn State University, USA).



**Acknowledgements**

Financial Support was provided by National Science Foundation Grant EAR–0725019 (C. Duffy), EAR–1239285 (S. Brantley), and EAR–1331726 (S. Brantley) for the Susquehanna Shale Hills Critical Zone Observatory.

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
