# Peer review of "Proposed Best Practices for Collaboration at Cross-disciplinary Observatories"

_Biogeosciences, 2019_

## Referee Comment (RC1) · Sybil Seitzinger (Referee) · 26 Aug 2019

The premise of this submission is sound: that large interdisciplinary science teams face a number of challenges that smaller, single domain teams often do not. Development of guidelines of best practices generally do not exist for interdisciplinary teams.

This paper is the manual used by a large interdisciplinary long-term field-based research effort. It could provide useful information for other similar teams to develop their own guidelines from. There are also aspects that could apply to very different, non-field, types of projects. For example 2.1.3. use of someone's code or model output. . ...", 2.1.8 how long should co-authors have to review manuscript", etc.

The current Perspective paper would need to be considerably shortened (perspectives

guidelines state such papers should be very short (a few pages)). This could be done by removing/summarizing some of the detail that is specific to this project – condense to key points. As one example, lines 265-295 are very specific to this particular project.

Are there social scientists involved in the project, or only biogeophysical? If the former, would have been useful to specifically discuss how to resolve some of the challenges integrating these often quite different research approaches, etc..

Given that this is an existing manual, it does not seem appropriate to comment, as a reviewer, on whether I agree with the specific guidelines they have established.

---

## Referee Comment (RC2) · Justus van Beusekom (Referee) · 30 Aug 2019

Interdisciplinary science is necessary to further our understanding of the earth system, but managing the scientific work of such projects is challenging especially in the light of the manifold interdependences. In the present ms the authors describe the present state of an ongoing document describing the best practice in a large interdisciplinary project (SSH CZO). In the present article, the 10 points are described including Authorship, Installing, Using and Removing Infrastructure, sharing data, project management, student advise, outreach and reporting.

I read the paper with great interest and recognised several of the issues.

A paper like this is somewhat unusual, especially, as the paper is based on a document that is meant to change (improve) based on internal discussions. However, given the theme, it certainly will be of general interest and of general importance for other interdisciplinary projects and for the scientists involved.

Therefore, I do support that the present ms will ultimately be published. However, I have suggestion that may be considered by the authors.

1) The author list includes the SSHCZO team. This has been done before, but I would welcome an addition to the section 2.1 that discusses the inclusion of teams as authors: e.g. how to document their contribution to a paper (as is expected in many papers). In what respect is the inclusion different from an acknowledgement?

2) Whereas Section 2.1 is generic, several of the further points (especially 2.2 -2.7) are quite specific to the SSHCZO. Is it possible to include /extract some kind of generic conclusions that may enhance the applicability of the present document to a wider public?

3) Maybe I overlooked it, but I would welcome link to the living document.

---

## Author Comment (AC2) · 14 Oct 2019

Reviewer comment 1: Interdisciplinary science is necessary to further our understanding of the earth system, but managing the scientific work of such projects is challenging especially in the light of the manifold interdependences. In the present ms the authors describe the present state of an ongoing document describing the best practice in a large interdisciplinary project (SSH CZO). In the present article, the 10 points are described including Authorship, Installing, Using and Removing Infrastructure, sharing data, project management, student advise, outreach and reporting. I read the paper with great interest and recognised several of the issues.

Author response 1: Thank you, we are glad the paper offered some generalizable

insights.

Reviewer comment 2: The author list includes the SSHCZO team. This has been done before, but I would welcome an addition to the section 2.1 that discusses the inclusion of teams as authors: e.g. how to document their contribution to a paper (as is expected in many papers). In what respect is the inclusion different from an acknowledgement?

Author response 2: Thank you for this suggestion. We agree that a discussion of how to define a team is appropriate to add to the manuscript. In response to this comment, we added section 2.1.15

Reviewer comment 3: Whereas Section 2.1 is generic, several of the further points (especially 2.2 -2.7) are quite specific to the SSHCZO. Is it possible to include /extract some kind of generic conclusions that may enhance the applicability of the present document to a wider public?

Author response 3: We agree that these sections included specificity that may not have been needed. In response to this review, we edited sections 2.2 -2.7 to speak more generally about permitting and best practices 2.2, 2.3, and 2.6. However, 2.2 received the most heavy handed editing. Our idea was that organizational structure like "steering committee" and "program manager" were in fact generic and relevant to all CZOs (lines 96-105). Thus, when we discuss our "steering committee" our hope is that readers see this not as a unique case, but rather as a general model for CZO organization structure. We changed the title of "Watershed Specialist" to "Field Operations Specialist" in recognition of the fact that all CZOs would have field operations, but they might not focus on watersheds.

Reviewer comment 4: Maybe I overlooked it, but I would welcome link to the living document.

Author response 4: Excellent point. We added the the link to the living document in section 3.

---

## Author Response (AR1)

Below please find the same review responses that we uploaded previously to the discussion in posts that were linked to each review. The accompanying manuscript file incorporates all of the changes below in yellow highlights and italics text.

Reviewer #1 (Sybil Seitzinger)

The premise of this submission is sound: that large interdisciplinary science teams face a number of challenges that smaller, single domain teams often do not. Development of guidelines of best practices generally do not exist for interdisciplinary teams.

This paper is the manual used by a large interdisciplinary long-term field-based research effort. It could provide useful information for other similar teams to develop their own guidelines from. There are also aspects that could apply to very different, non-field, types of projects. For example 2.1.3. use of someone's code or model output. . ...", 2.1.8 how long should co-authors have to review manuscript", etc.

*Thank you, we hoped this would be the case.*

The current Perspective paper would need to be considerably shortened (perspectives C1 BGD Interactive comment Printer-friendly version Discussion paper guidelines state such papers should be very short (a few pages)). This could be done by removing/summarizing some of the detail that is specific to this project – condense to key points.

*We had contacted the editors about the length of our manuscript prior to submission. The editors had agreed that despite the unorthodox length, the manuscript would be considered. If the editors feel the paper should be shortened to a few pages we would need to consider if that is possible. Thus, we did not endeavor to shorten the overall length of the manuscript. However, we did change the paper length slightly in response to the next comment.*

As one example, lines 265-295 are very specific to this particular project. Are there social scientists involved in the project, or only biogeophysical? If the former, would have been useful to specifically discuss how to resolve some of the challenges integrating these often quite different research approaches, etc..

*We agree that these lines became too specific to the project. There are no social scientists involved in our project, though this seems like an area for future improvement of this living document.*

*To address the high level of specificity, we reworked this section (former lines 265-295), gearing it toward a more general audience, and in doing so cut the length of the text significantly. We also removed some details about permitting from section 1 that were redundant with section 2.2. We did not change the document to include social science perspectives because we lack the experience to do so.*

Given that this is an existing manual, it does not seem appropriate to comment, as a reviewer, on whether I agree with the specific guidelines they have established.

*No changes were made to the document with respect to this comment.*

Reviewer #2 (Justus van Beusekom)

Interdisciplinary science is necessary to further our understanding of the earth system, but managing the scientific work of such projects is challenging especially in the light of the manifold interdependences. In the present ms the authors describe the present state of an ongoing document describing the best practice in a large interdisciplinary project (SSH CZO). In the present article, the 10 points are described including Authorship, Installing, Using and Removing Infrastructure, sharing data, project management, student advise, outreach and reporting.

I read the paper with great interest and recognised several of the issues.

*Thank you, we are glad the paper offered some generalizable insights.*

A paper like this is somewhat unusual, especially, as the paper is based on a document that is meant to change (improve) based on internal discussions. However, given the theme, it certainly will be of general interest and of general importance for other interdisciplinary projects and for the scientists involved.

Therefore, I do support that the present ms will ultimately be published. However, I have suggestion that may be considered by the authors.

1) The author list includes the SSHCZO team. This has been done before, but I would welcome an addition to the section 2.1 that discusses the inclusion of teams as authors: e.g. how to document their contribution to a paper (as is expected in many papers). In what respect is the inclusion different from an acknowledgement?

*Thank you for this suggestion.  We agree that a discussion of how to define a team is appropriate to add to the manuscript.*

*In response to this comment, we added section 2.1.15*

2) Whereas Section 2.1 is generic, several of the further points (especially 2.2 -2.7) are quite specific to the SSHCZO. Is it possible to include /extract some kind of generic conclusions that may enhance the applicability of the present document to a wider public?

*We agree that these sections included specificity that may not have been needed.*

*In response to this review, we edited sections 2.2 -2.7 to speak more generally about permitting and best practices 2.2, 2.3, and 2.6.  However, 2.2 received the most heavy handed editing.  Our idea was that organizational structure like "steering committee" and  "program manager" were in fact generic and relevant to all CZOs (lines 96-105).  Thus, when we discuss our "steering committee" our hope is that readers see this not as a unique case, but rather as a general model for CZO organization structure. We changed the title of "Watershed Specialist" to "Field Operations Specialist" in recognition of the fact that all CZOs would have field operations, but they might not focus on watersheds.*

3) Maybe I overlooked it, but I would welcome link to the living document.

Excellent point.

We added the the link to the living document in section 3.